# Decomposing Out-of-Distribution Error in Conditional Flow Matching via Wasserstein Geometry

**Long HC Pham** [1]

## Abstract

Conditional flow matching has emerged as a powerful generative modeling framework that learns a vector field to transport an initial distribution toward a target data distribution. However, theoretical understanding of its out-of-distribution (OOD) performance under unseen conditions remains limited. In this work, we establish a rigorous geometric formulation to decompose the source of generalization error. We treat the conditional task as a map from the condition space to the Wasserstein space and derive a generalization bound under a coarse embedding assumption. The resulting decomposition separates OOD error into three interpretable components: *Interpolation Sparsity*, *Geometric Distortion*, and *In-Distribution Fit*. Our empirical evaluation confirms that this framework demonstrates three key functions: (1) it acts as a diagnostic tool that tracks the dynamics of generalization during training; (2) it identifies dataset-specific failure modes (e.g., topological gaps, geometric instability); and (3) it enables mathematically motivated interventions that yield predictable gains by minimizing specific terms.

## 1. Introduction

Flow-based generative models construct data distributions by learning vector fields that transport a simple base measure to a target distribution along continuous paths (Lipman et al., 2023). Conditional flow matching extends this idea to families of conditional distributions $\{p^*(\cdot \mid c)\}_{c \in \mathcal{C}}$, enabling controllable generation (Fetaya et al., 2020; Tong et al., 2024). Yet, the theoretical characterization of generalization to unseen conditions remains incomplete. In practice, empir-

ical performance often depends on inference-time heuristics inherited from diffusion models, most notably classifier-free guidance (Tong et al., 2024). When applied within the vector field formulation, however, such heuristics lack formal guarantees and may induce undesirable behaviors, including norm amplification and mode collapse. Recent theoretical work further suggests that, in the absence of these interventions, the standard flow matching objective admits a kernel regression (Albergo & Vanden-Eijnden, 2023), which biases the learned dynamics toward mean-reverting behavior and can lead to geometric and invertibility limitations in high-dimensional out-of-distribution regimes (Kirichenko et al., 2020; Liu et al., 2022).

From a geometric perspective, each conditional distribution $p^*(\cdot \mid c)$ can be identified with a point in the Wasserstein space $(\mathcal{P}_2(\mathbb{R}^d), W_2)$, where $\mathcal{P}_2(\mathbb{R}^d)$ denotes the set of probability measures on $\mathbb{R}^d$ with finite second moment and $W_2$ the associated 2-Wasserstein distance. Under this notation, conditional generative modeling corresponds to learning a map

$$f : \mathcal{C} \to (\mathcal{P}_2(\mathbb{R}^d), W_2)$$

where generalization to unseen conditions amounts to evaluating the learned map on inputs outside the observed subset of $\mathcal{C}$. Crucially, $(\mathcal{P}_2(\mathbb{R}^d), W_2)$ is a curved metric space with a nonlinear geodesic structure (Villani, 2008; Ambrosio et al., 2008), suggesting that naive interpolation or extrapolation across conditions may be fundamentally misaligned with the underlying geometry.

**Embedding Obstructions and Distortion.** While certain structured submanifolds (e.g., Gaussian families) admit isometric embeddings into Wasserstein space (Villani, 2008), $\mathcal{P}_2(\mathbb{R}^d)$ contains snowflake-type subsets that cannot be represented as the bi-Lipschitz image of any Euclidean domain (Andoni et al., 2018). As a consequence, standard Lipschitz constraints can be overly restrictive when modeling complex distributions. To address this, we adopt coarse embeddings (Pritchard & Weighill, 2024), which provide a principled justification for conditional flows by preserving relational structure while allowing unavoidable local additive distortions. Formal proofs are deferred to Appendix B.2.

Generalization to an unseen condition $\bar{c}$ can be analyzed

---

[1]University of Engineering and Technology, Vietnam National University, Hanoi, Vietnam. Correspondence to: Long HC Pham <htaevd@gmail.com>.

*Proceedings of the 43rd International Conference on Machine Learning*, Seoul, South Korea. PMLR 306, 2026. Copyright 2026 by the author(s).

through a Wasserstein barycentric construction over nearby training conditions $\{c_i\}_{i=1}^N$:

$$\bar{\mu} \; = \; \arg\min_{\mu \in \mathcal{P}_2(\mathbb{R}^d)} \sum_{i=1}^N w_i \, W_2^2(\mu, \, p^*(\cdot \mid c_i)),$$

where the weights $w_i$ encode the relationship between $\bar{c}$ and the observed conditions. The model then produces $\hat{p}(\cdot \mid \bar{c})$ by integrating the learned conditional vector field.

Our main contribution is a generalization bound that decomposes

$$W_2(\hat{p}(\cdot \mid \bar{c}), \, p^*(\cdot \mid \bar{c}))$$

into three interpretable components: (i) a Wasserstein barycentric approximation term, (ii) a coarse distortion term capturing geometric mismatch between condition space and Wasserstein space, and (iii) an in-distribution estimation term reflecting finite sample and model fitting error. This decomposition characterizes when conditional flows generalize and when they fail, providing a geometry-aware alternative to Lipschitz-based analyses.

**Scope of Contribution.** Unlike approaches that modify the training objective or vector field parameterization to enforce optimal transport structure (Pooladian et al., 2023; Tong et al., 2024; Fetaya et al., 2020; Albergo & Vanden-Eijnden, 2023), our work accepts the trained vector field as given and provides a post-hoc geometric diagnostic for its OOD limits. A central premise of our analysis is quantifying OOD behavior relative to the *Wasserstein geodesic interpolation* (displacement interpolation) (McCann, 1997). We validate this framework through controlled and high-dimensional settings, demonstrating a causal link between geometric stability and generalization bounds. In these settings, the proposed decomposition functions as a white-box diagnostic, differentiating between topological and geometric sources of OOD failure in ways that heuristic evaluations cannot.

**Relation to Conditional and Wasserstein Dynamics.** While our framework accepts the trained vector field as given, recent developments in the field focus on modifying the training objective to respect specific conditional geometries. Approaches based on *conditional Wasserstein distances* (Carlier et al., 2016; Muzellec & Cuturi, 2018) aim to minimize distances between conditional transport plans directly, thereby enforcing structure on the joint distribution rather than on marginal transport maps alone. Similarly, recent advances in flow matching on general geometries (Chen & Lipman, 2024) and metric flow matching on the space of measures (Kapusniak et al., 2024) seek to constrain dynamics to approximate geodesic paths on the underlying manifold. In the context of explicit condition handling, the literature on *Wasserstein regression* and *barycentric extrapolation* (Bigot & Klein, 2018; Chen et al., 2023) typically

models the mapping

$$c \; \mapsto \; \mu_c \in \mathcal{P}_2(\mathbb{R}^d)$$

as a Fréchet-type regression problem. However, these methods often rely on implicit regularity assumptions, such as bi-Lipschitz continuity of transport maps (Mi et al., 2020). In contrast, our work explicitly analyzes regimes in which these smoothness assumptions fail, through the lens of coarse embedding obstructions and the resulting error components that arise under out-of-distribution (OOD) evaluation.

## 2. Background and Geometric Foundations

### 2.1. Conditional Flow Matching

We adopt the conditional flow matching (CFM) framework (Lipman et al., 2023; Tong et al., 2024). Given a data distribution $\rho_1(x|c)$ conditioned on $c \in \mathcal{C}$, CFM learns a time-dependent vector field $v_\theta(t, x|c)$ transporting a source $\rho_0$ to $\rho_1$ via the ODE $\frac{dx_t}{dt} = v_\theta(t, x_t)$. The density evolves according to the continuity equation:

$$\frac{\partial \rho_t}{\partial t} + \nabla \cdot (v_\theta(t, \cdot|c)\, \rho_t) = 0. \qquad (1)$$

Training minimizes the expected regression loss between $v_\theta$ and a ground-truth field constructed via optimal transport conditional couplings.

### 2.2. Wasserstein Geometry and Otto Calculus

Following Otto (2001), the space $(\mathcal{P}_2(\mathbb{R}^d), W_2)$ admits a formal infinite-dimensional Riemannian structure. The tangent space at $\mu$ is given by

$$T_\mu \mathcal{P}_2 = \{\nabla \phi : \phi \in L_\mu^2\},$$

equipped with the metric

$$g_\mu(v_1, v_2) = \int_{\mathbb{R}^d} \langle v_1(x), v_2(x) \rangle \, d\mu(x).$$

Under this geometric view, conditional flow matching learns a curve $\{\mu_t\}_{t \in [0,1]}$ intended to approximate the Wasserstein geodesic (displacement interpolation) between source and target distributions. The regularity of this Riemannian structure — specifically the smoothness of the tangent bundle and the existence of well-defined gradient flows — underpins the Lipschitz continuity assumptions we impose on the learned and ground-truth velocity fields in Section 3.3.

### 2.3. Coarse Geometry and Embedding Obstructions

A critical theoretical hurdle in conditional modeling is the geometric mismatch between the condition space $\mathcal{C}$ (typically Euclidean $\mathbb{R}^n$, a "Type 2" space in the sense of Banach space geometry) and the target Wasserstein space $\mathcal{P}_2(\mathcal{X})$.

$(\mathcal{P}_2(\mathbb{R}^d), W_2)$ is known to be *snowflake-universal* (Andoni et al., 2018; Pritchard & Weighill, 2024). A key result in metric geometry states that snowflake-universal spaces contain subsets that cannot be represented as the bi-Lipschitz image of any *Type-2* space (i.e., any Euclidean or Hilbert space, in the sense of Banach space geometry (Andoni et al., 2018)); any map $f : \mathcal{C} \to \mathcal{P}_2$ attempting bi-Lipschitz fidelity must introduce unbounded non-linear distortion. This implies that assuming standard Lipschitz continuity for the mapping $c \mapsto \mu_c$ is mathematically unjustified in general for complex distributions. To address this, we relax strict bi-Lipschitz assumptions to *coarse embeddings* (Pritchard & Weighill, 2024). A map $f : \mathcal{C} \to \mathcal{P}_2$ is a coarse embedding if there exist non-decreasing functions $\rho_-, \rho_+$ such that

$$\rho_-\big(d_{\mathcal{C}}(c_i, c_j)\big) \leq W_2\big(f(c_i), f(c_j)\big) \leq \rho_+\big(d_{\mathcal{C}}(c_i, c_j)\big).$$

where $d_{\mathcal{C}}$ denotes the metric on the condition space (e.g., the Euclidean distance when $\mathcal{C} = \mathbb{R}^n$). Specifically, we assume a *quasi-isometric* relationship with additive slack:

$$a\, d_{\mathcal{C}}(c_i, c_j) - b \leq W_2(\mu_{c_i}, \mu_{c_j}) \leq A\, d_{\mathcal{C}}(c_i, c_j) + B \quad (2)$$

with $a, A > 0$ and $b, B \geq 0$. This preserves global topology while tolerating local metric distortions caused by the snowflake geometry of $\mathcal{P}_2$, providing a sound basis for our bounds.

## 2.4. Generalization via Barycenters

For OOD generalization, we idealize the target distribution for an unseen $\bar{c}$ as a **Wasserstein Barycenter** of training conditions $\{c_i\}$:

$$\bar{\mu} = \arg \min_{\nu \in \mathcal{P}_2} \sum_{i=1}^{N} w_i W_2^2(\nu, \mu_{c_i}), \quad (3)$$

where $w_i \geq 0$ and $\sum_{i=1}^{N} w_i = 1$. We emphasize that $\bar{\mu}$ serves as a *geometric reference distribution* for $\bar{c}$, not a ground-truth target; the bound is therefore relative to this barycentric reference, and grows interpretably when the true conditional distribution lies outside the convex hull of training conditions.

## 3. Methods

Building on the geometric background in Section 2, including Wasserstein geometry, Otto's Riemannian formulation, and coarse embedding theory, we develop a framework for analyzing out-of-distribution (OOD) generalization in conditional flow models. Our approach proceeds in three steps:

1. We formulate conditional generation as a learned map $f : \mathcal{C} \to \mathcal{P}_2(\mathcal{X})$, which assigns each condition $c \in \mathcal{C}$ to a target measure $\mu_c = f(c)$ in Wasserstein space.

2. For an unseen condition, we define a reference target distribution via Wasserstein barycentric interpolation of neighboring conditional measures in $\mathcal{P}_2(\mathcal{X})$.

3. We derive an error decomposition that explicitly isolates the effect of geometric distortion induced by the learned map $f$.

This distortion is quantified through a *coarse embedding* assumption. As shown in Appendix B, such an assumption is not a technical convenience but a mathematical necessity: stricter Wasserstein Lipschitz assumptions are incompatible with known embedding obstructions of $(\mathcal{P}_2, W_2)$ and fail for complex data distributions.

### 3.1. Problem Setup: Conditional Flow Matching

We begin with a conditional flow matching framework, as defined in the preliminaries, which learns a time-dependent vector field $v_\theta(x, t \mid c)$ that transports a source distribution $\rho_0(x)$ to a target distribution $\rho_1(x \mid c)$ conditioned on $c \in \mathcal{C}$, where $\mathcal{X} \subseteq \mathbb{R}^d$ denotes the data space and $\mathcal{C}$ the condition space. Identifying $\rho_t(\cdot \mid c)$ with its associated probability measure, the learned flow satisfies the continuity equation

$$\frac{\partial \rho_t(x \mid c)}{\partial t} + \nabla \cdot \big(\rho_t(x \mid c)\, v_\theta(x, t \mid c)\big) = 0.$$

**Sampling and Conditional Structure.** Each training pair $(x, c)$ is drawn from a joint distribution $\rho_1(x, c) = \rho_1(x \mid c)\, p(c)$, where $p(c)$ defines the sampling measure over the condition space $\mathcal{C}$. For example, when $\mathcal{C}$ represents class labels, samples are generated by first drawing $c \sim p(c)$ and then $x \sim \rho_1(x \mid c)$. The conditional vector field $v_\theta(x, t \mid c)$ therefore governs both the temporal transport in data space $\mathcal{X}$ and its dependence on the conditioning variable $c$, a structure whose geometric implications we analyze in subsequent sections.

**Geometric Conditioning.** Unlike simple categorical conditioning (e.g., one-hot labels), we view the condition space $\mathcal{C}$ as a geometric manifold equipped with a metric $d_{\mathcal{C}}$. This formulation enables reasoning about **interpolation and extrapolation** between conditions: given observed conditions $c_1, c_2, \ldots, c_n$, an unseen condition $\bar{c}$ occupies a location in $\mathcal{C}$ defined by its metric relations under $d_{\mathcal{C}}$. The conditional flow matching model is therefore expected to generate $\rho_1(x \mid \bar{c})$ by leveraging the learned structure of $v_\theta$ across nearby conditions, even when $\bar{c}$ lies beyond the convex hull of the training set.

**Challenge of Out-of-Distribution (OOD) Conditioning.** Standard formulations of conditional FM assume that conditioning operates over discrete or flat Euclidean spaces, where geometric relations between conditions are not explicitly encoded. As a result, when an unseen condition $\bar{c}$

is presented, the model often fails to generalize smoothly, since it lacks a meaningful notion of continuity or curvature in $\mathcal{C}$. This limitation motivates introducing a **geometric relaxation** of the condition space, where mappings between $\mathcal{C}$ and the data manifold in $\mathcal{X}$ are treated through their underlying transport geometry.

In the following, we formalize this geometric perspective through the lens of the Wasserstein space, which provides a principled way to endow the space of probability measures with a Riemannian structure.

### 3.2. Geometric Modeling of the Conditional Map

Building on the Riemannian interpretation of $\mathcal{P}_2(\mathbb{R}^d)$ outlined in Section 2, we cast the conditional generative task not merely as function approximation, but as learning a manifold-valued map

$$f : \mathcal{C} \to (\mathcal{P}_2(\mathbb{R}^d), W_2), \quad c \mapsto \mu_c.$$

In this framework, the learned vector field $v_\theta$ defines the tangent vectors that transport the base measure along the geodesic path toward $f(c)$. Consequently, generalization to an unseen condition $\bar{c}$ corresponds to extrapolating this map $f$ to a new point on the Wasserstein manifold.

To rigorously evaluate this extrapolation, we rely on the **Wasserstein Barycenter** (Eq. (3)) as the geometric reference distribution. Specifically, if the map $f$ is smooth (respecting the coarse embedding structure), the distribution at an unseen $\bar{c}$ should be well-approximated by the barycenter of its neighbors in condition space. This geometric proxy allows us to formally decompose the OOD error into model-specific failures versus fundamental geometric obstructions.

### 3.3. Geometric Structure and Key Assumptions

The bi-Lipschitz continuity of the conditional map $c \mapsto \mu_c$ with respect to $d_\mathcal{C}$ and $W_2$ is a natural but generally invalid assumption. While finite-dimensional condition spaces are Type-2 (Euclidean-like), $(\mathcal{P}_2(\mathcal{X}), W_2)$ is snowflake-universal (Andoni et al., 2018). This means it contains fractal subsets that cannot be represented as the bi-Lipschitz image of any Euclidean domain. Consequently, strictly enforcing Lipschitz continuity prevents the model from capturing these complex geometries. To address this incompatibility, we replace the bi-Lipschitz constraint with a *coarse embedding* assumption (Pritchard & Weighill, 2024):

$$a\, d_\mathcal{C}(c_1, c_2) - b \ \leq \ W_2(\mu_{c_1}, \mu_{c_2}) \ \leq \ A\, d_\mathcal{C}(c_1, c_2) + B,$$
$$(a, A > 0; \ b, B \geq 0).$$

This preserves global relational structure while allowing controlled additive distortion. This relaxation is sufficient (Appendix B) and avoids the geometric incompatibilities that invalidate strict Wasserstein–Lipschitz assumptions.

We additionally assume that the ground-truth $v^*$ and learned flow fields $v_\theta$ are Lipschitz with respect to their conditional dependence. Viewing the vector field as a functional of the condition, this assumption takes the form

$$\|v_\theta(\cdot, t \mid c_1) - v_\theta(\cdot, t \mid c_2)\| \ \leq \ L_c\, d_\mathcal{C}(c_1, c_2),$$

uniformly in $t$. While strict global Lipschitz continuity is difficult to guarantee in neural networks, existing flow matching theory (Lipman et al., 2023) relies on analogous regularity assumptions to ensure existence and uniqueness of the induced ODE solutions. We therefore adopt this condition as a standard tractability assumption, ensuring stability of the learned dynamics once geometric distortion is accounted for.

Together, these assumptions make a feasible and realistic structure for bounding generalization error across unseen conditions.

### 3.4. Error Bound for Unseen Conditions

Let $\bar{c}$ be an unseen condition with ground-truth distribution $\mu_{\bar{c}} = p^*(\cdot \mid \bar{c})$. We analyze the generalization error by decomposing the trajectory of the learned flow relative to the training conditions $\{c_i\}_{i=1}^N$.

Assume that both the learned vector field $v_\theta$ and the ground truth $v^*$ are Lipschitz in the state variable $x$ (with constant $L_x$) and in the conditional dependence (with constants $L_c$ and $L_c^*$ respectively). Standard stability results for continuity equations yield, via Grönwall's inequality:

$$W_2(\hat{p}(\cdot \mid \bar{c}), p^*(\cdot \mid \bar{c})) \ \leq \ e^{L_x^*} \int_0^1 \mathcal{E}(\bar{c}, t)\, dt,$$

where $\mathcal{E}(\bar{c}, t) = \sup_x \|v_\theta(x, t \mid \bar{c}) - v^*(x, t \mid \bar{c})\|$ denotes the instantaneous velocity approximation error at condition $\bar{c}$ and time $t$.

We introduce a *barycentric decomposition* using weights $\{w_i\}_{i=1}^N$. By the triangle inequality, for any fixed $(x, t)$:

$$\mathcal{E}(\bar{c}, t) \ \leq \ \sum_{i=1}^N w_i \Big[ \|v_\theta(x, t \mid \bar{c}) - v_\theta(x, t \mid c_i)\|$$
$$+ \ \|v_\theta(x, t \mid c_i) - v^*(x, t \mid c_i)\|$$
$$+ \ \|v^*(x, t \mid c_i) - v^*(x, t \mid \bar{c})\| \Big].$$

The second term corresponds to the weighted training error, bounded by $\epsilon_{\text{train}} = \max_i \int \|v_\theta - v^*\| dt$. For the first and third terms, we apply the Lipschitz stability of the vector fields with respect to the condition $c$:

$$\|v_\theta(\cdot \mid \bar{c}) - v_\theta(\cdot \mid c_i)\| + \|v^*(\cdot \mid c_i) - v^*(\cdot \mid \bar{c})\|$$
$$\leq (L_c + L_c^*)\, d_\mathcal{C}(c_i, \bar{c}).$$

Finally, applying the *coarse embedding assumption* (Eq. (2)) allows us to replace the Euclidean condition distance $d_{\mathcal{C}}$ with the transport distance:

$$d_{\mathcal{C}}(c_i, \bar{c}) \leq \frac{1}{a}\big(W_2(\mu_{c_i}, \mu_{\bar{c}}) + b\big).$$

**Final Generalization Bound.** Let $K = \frac{e^{L_x^*}(L_c + L_c^*)}{a}$, $\delta = \frac{e^{L_x^*}(L_c + L_c^*)b}{a}$, and $C = e^{L_x^*}$. Combining the terms yields:

$$W_2(\hat{p}(\cdot \mid \bar{c}), \mu_{\bar{c}}) \leq \underbrace{K \sum_{i=1}^{N} w_i\, W_2(\mu_{c_i}, \mu_{\bar{c}})}_{\text{(I) Interpolation Sparsity}} \qquad (4)$$
$$+ \underbrace{\delta}_{\text{(II) Distortion}} + \underbrace{C\,\epsilon_{\text{train}}}_{\text{(III) In-Dist. Fit}} .$$

**Geometric Interpretation.** Equation (4) decomposes the OOD error into three interpretable components:

1. **Interpolation Sparsity (I):** Measures the Wasserstein distance from the training support to the target $\mu_{\bar{c}}$. This term is minimized when $\mu_{\bar{c}}$ lies in the geometric center (barycenter) of the training distributions, and grows as the target drifts away from the training hull.

2. **Geometric Distortion (II):** A strictly positive constant derived from the "snowflake" nature of $\mathcal{P}_2$. It represents the unavoidable minimum error incurred by embedding a flat condition space into a curved Wasserstein manifold.

3. **In-Distribution Fit (III):** The standard approximation error on the training set, amplified by the Lipschitz constant of the dynamics.

**Selection of Weights.** To minimize the bound, the weights $w_i$ should concentrate on training points geometrically closest to $\bar{c}$. We instantiate these using a softmax kernel:

$$w_i(\bar{c}) = \frac{\exp\big(-\frac{1}{\tau}d_{\mathcal{C}}(\bar{c}, c_i)\big)}{\sum_{j=1}^{N} \exp\big(-\frac{1}{\tau}d_{\mathcal{C}}(\bar{c}, c_j)\big)}. \qquad (5)$$

This formulation naturally adapts to both interpolation (where weights effectively average neighbors) and extrapolation regimes.

**Implications for Extrapolation.** A key insight from Eq. (4) is the behavior of the sparsity term (I) under linear shifts. When $\bar{c}$ moves away from the training set (extrapolation), $d_{\mathcal{C}}(\bar{c}, c_i)$ grows linearly. Due to the coarse embedding property ($W_2 \approx a \cdot d_{\mathcal{C}}$), the term $\sum w_i W_2$ also grows linearly. This theoretically predicts that **OOD error scales linearly with geometric drift**, a phenomenon we verify in Section 4.1.

## 4. Experiments

### 4.1. Exact Validation on Controlled Manifolds

A primary challenge in verifying theoretical generalization bounds on datasets is the intractability of the true Wasserstein distance $W_2(p, q)$ and the reliance on potentially lossy latent proxies. Recent reviews correctly note that verifying constants such as Lipschitz bounds on high-dimensional inputs is nearly impossible. To address this limitation and rigorously validate (4), we first evaluate our framework on a controlled low-dimensional manifold where ground-truth distributions are known analytically and exact Optimal Transport distances are computationally tractable.

**Dataset: Linear Translation.** We construct a synthetic dataset $x \in \mathbb{R}^2$ based on the "two moons" manifold, which undergoes a linear translation parametrized by a scalar condition $c$. The ground truth follows $\mu_c^* = \mathbf{x}_{\text{base}} + c \cdot \vec{v}$, with drift velocity $\vec{v} = [0.5, 0.5]^\top$. The model is trained on conditions $c \in [0, 4.0]$ and evaluated in the strict extrapolation regime $c \in (4.0, 7.0)$. This setup isolates the **Intrinsic Data Drift** (Term 1, Sparsity): since the true manifold moves linearly, the optimal transport cost increases linearly with $c$. Any error exceeding this linear growth corresponds to **Geometric Distortion** ($\delta$) caused by model instability.

**Modeling Extrapolation.** In the extrapolation regime ($c \in (4.0, 7.0)$), the true manifold continues to translate linearly. However, the standard softmax weights (Eq. (5)) constrain the model's barycentric reference to the nearest training neighbors $c_{train} \in [0, 4.0]$. This creates a geometric discrepancy between the *true* extrapolated target and the *interpolated* reference expected by the model. Our bound captures this via the **Geometric Distortion term** ($T_2$). In extrapolation, the transport cost $W_2(f(\bar{c}), f(c_i))$ grows linearly; if the model fails to maintain the coarse embedding constants, the distortion $\delta$ explodes. The robust model is designed to minimize this by enforcing a tighter Lipschitz constant $L_c$, effectively forcing the linear trend to continue outside the convex hull.

**Model Comparison and Results.** We compare a **Baseline** flow model (unconstrained, using high-frequency Fourier embeddings to simulate overfitting) against a **Robust** model designed to minimize the Lipschitz constant $L_c$ via spectral normalization and a VAE bottleneck on the condition embedding. Figure 1 summarizes the results. The robust model strictly respects the theoretical upper bound, exhibiting monotonic error growth parallel to the irreducible intrinsic data drift, while the baseline oscillates and frequently exceeds the bound due to high geometric sensitivity. Estimating the empirical expansion constant ($K_{emp} \approx 10$) at the onset of OOD confirms that extrapolation error grows linearly with distance from the training support, as predicted by Eq. (4). The corresponding error decomposition further

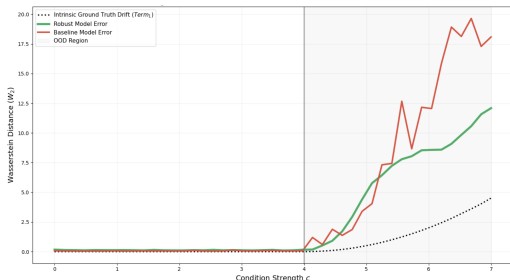

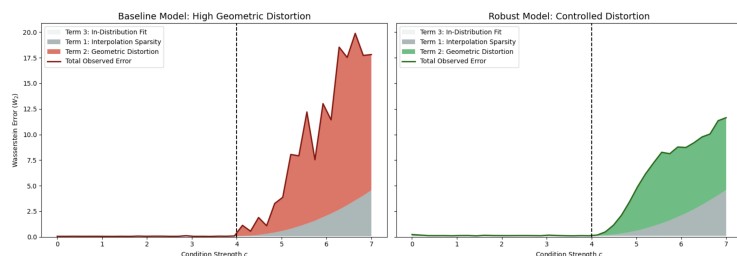

*(a)* **Verification of the bound.** The **black dotted** line represents the irreducible *intrinsic data drift* ($T_1$). The **robust model (green)** strictly respects the theoretical upper bound (green dashed), exhibiting monotonic error growth parallel to the drift. The **baseline (red)** oscillates wildly, exceeding the bound due to high geometric sensitivity.

*(b)* **Error decomposition.** Stacked plots visualizing the three error terms from (4). For the baseline (left), the error is dominated by **geometric distortion** ($T_2$, red area). For the robust model (right), this distortion is effectively suppressed, leaving the error proportional to the interpolation sparsity ($T_1$, grey).

*Figure 1.* **Exact validation on linear drift manifold.** These results confirm that spectral regularization converts the high-variance failure modes of standard models into predictable linear drift.

reveals that this performance gap is driven entirely by the **Geometric Distortion** term ($T_2$): the robust formulation effectively suppresses this instability, leaving the error dominated by interpolation sparsity ($T_1$), thereby demonstrating that improved generalization arises from stabilizing the map $f : \mathcal{C} \to \mathcal{P}_2$, rather than altering the data distribution.

## 4.2. High-Dimensional Diagnostics

Having validated the theoretical bounds on exact manifolds, we now scale our analysis to image benchmarks: **MNIST**, **CIFAR-10**, and **CelebA**. Our goal here is *diagnostic*: using the derived decomposition to identify whether a model is failing due to topology (Term 1) or stability (Term 2) in (4).

### 4.2.1. EXPERIMENTAL SETUP

Since the true conditional distribution $f(\bar{c})$ is unknown for images, we use the **latent Wasserstein distance** (computed in the VAE latent space $\mathcal{Z}$) as a proxy for the true transport distance. We track the components of (4):

- **interpolation error** ($\mathcal{E}_{\text{interp}}$): quantifies data coverage / sparsity ($T_1$) by measuring the Wasserstein distance between an unseen condition and the convex hull of training latents.

- **geometric distortion proxy** ($\hat{\delta}$): quantifies stability ($T_2$) using the local condition number of the generator, computed via finite-difference perturbations in latent space.

**Proxy evaluation of error terms.** While the theoretical bound depends on weights defined via the condition metric $d_{\mathcal{C}}$, this metric is not directly observable for discrete image classes. To obtain a diagnostic estimate of the interpolation

sparsity term, we instead optimize the interpolation weights to minimize the Wasserstein distance to the target distribution. This yields a tight lower bound on the topological error: failure under this optimized interpolation certifies the absence of a valid geometric path, independent of the weighting rule. Implementation details are provided in Appendix A.

### 4.2.2. VALIDATION OF BOUND TIGHTNESS ON IMAGES

In Figure 2, we track the theoretical bound (purple dotted) alongside the observed out-of-distribution mean squared error (black solid). Two key behaviors demonstrate that the proxy decomposition tracks the structural predictions of the bound:

1. **Sensitivity to instability:** in CelebA (Figure 2c), fluctuations in the observed MSE around epoch 100 are immediately reflected by the bound, indicating that the proxy $\hat{\delta}$ captures local generator roughness.

2. **Structural alignment:** the bound closely follows the learning trajectory across datasets, supporting the interpretation that OOD error arises jointly from the topological position of the test sample (sparsity) and the instantaneous geometric stability of the model (distortion).

### 4.2.3. DECOMPOSING FAILURE MODES

The decomposition (Figure 2, top) enables a diagnosis of how and why generalization fails across different datasets:

- **CIFAR-10 (coupled failure):** the model exhibits a "diagonal descent" trajectory (Figure 2h). The complexity of the image manifold induces difficulty in both

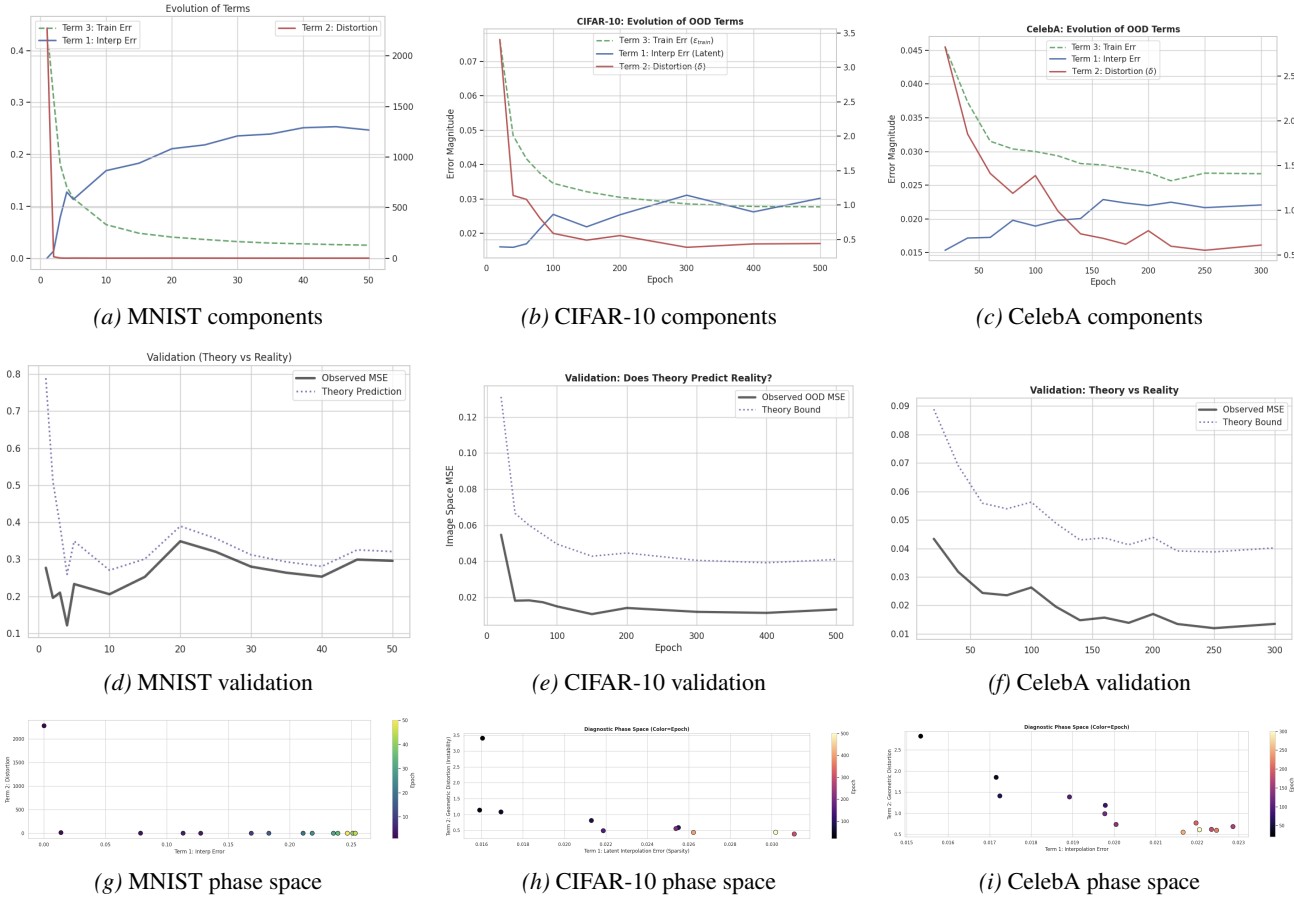

*Figure 2.* Visualization of model behavior across datasets. **(a–c)** Decomposition of theoretical error terms. **(d–f)** Comparison between generalization bound and true OOD deviation. **(g–i)** Phase space trajectories showing dataset-specific error modes.

identifying a coherent latent neighborhood (sparsity) and maintaining geometric stability (distortion). As a result, generalization is jointly constrained by topological coverage and geometric sensitivity.

- **CelebA (geometric failure):** the underlying semantic manifold (e.g., attributes such as hair color or facial structure) is relatively smooth, making topological interpolation straightforward (low $T_1$). Failure is instead dominated by **geometric distortion**, with $T_2$ remaining high early in training (Figure 2c), producing a "vertical descent" trajectory.

- **MNIST (topological failure):** geometric distortion is negligible, with $T_2$ near zero throughout training. However, the discrete class structure induces separated latent clusters, creating a "latent hole" that yields high interpolation error and results in **horizontal stagnation**. This identifies the failure mode as purely topological.

### 4.3. Empirical Validation of the Geometric Distortion Proxy

A key challenge in applying our theoretical bound (4) is that the geometric distortion term, $\delta$, depends on the implicit coarse embedding constants $(a, b)$ of the data manifold, which are intractable to compute for high-dimensional image data. We performed a calibration analysis on the CelebA dataset to verify structural consistency between our observable heuristic $\hat{\delta}$ and the true theoretical distortion term.

**Isolation of Geometric Residual.** To isolate the effect of geometric distortion from data sparsity and model fitting error, we define the *Observed Geometric Residual* ($\hat{E}_{\mathrm{gap}}$). While the theoretical bound (Eq. (4)) includes multiplicative expansion constants ($K, C \geq 1$), we analyze the unnormal-

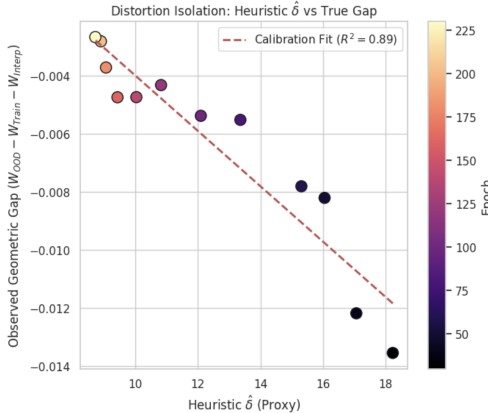

*Figure 3.* **Calibration of the Distortion Heuristic on CelebA.** We verify the relationship between our practical heuristic $\hat{\delta}$ (x-axis) and the isolated True Geometric Gap (y-axis, computed via Eq. (6)) throughout training. The strong linear correlation ($R^2 = 0.89$) demonstrates structural consistency between $\hat{\delta}$ and the observed geometric residual, supporting its use as a diagnostic proxy.

ized residual to track relative trends in the unexplained error:

$$\hat{E}_{\mathrm{gap}}(t) = W_2(\hat{p}_{\bar{c}}, p_{\bar{c}}^*)$$
$$- \left( \underbrace{\sum_i w_i W_2(\hat{p}_{c_i}, p_{c_i}^*)}_{\text{In-Dist. Train Error}} + \underbrace{\sum_i w_i W_2(p_{c_i}^*, p_{\bar{c}}^*)}_{\text{Interp. Sparsity}} \right). \tag{6}$$

Here, $\hat{E}_{\mathrm{gap}}$ serves as an observable proxy for the theoretical distortion term $\delta$. If our theory holds, this residual term must correlate strongly with the local curvature of the learned map.

**Computation of the Heuristic $\hat{\delta}$.** We estimate the proxy distortion $\hat{\delta}$ via the local Jacobian sensitivity of the conditional flow at the barycentric point. For a latent coordinate $z$ corresponding to the unseen condition $\bar{c}$, we apply a Gaussian perturbation $\mathcal{N}(0, \sigma^2 I)$ and measure the expansion:

$$\hat{\delta}(z) \approx \mathbb{E}_\epsilon \left[ \frac{\|D_\theta(z + \epsilon) - D_\theta(z)\|_2}{\|\epsilon\|_2} \right], \tag{7}$$

where $D_\theta$ is the trained decoder/flow. This effectively probes the local smoothness of the coarse embedding without requiring global Lipschitz constants.

**Results and Interpretation.** As shown in Figure 3, we observe a strong linear correlation ($R^2 = 0.89$) between the heuristic $\hat{\delta}$ and the observed $E_{\mathrm{gap}}$ throughout the training dynamics. The relationship demonstrates that as the model stabilizes (reducing local sensitivity $\hat{\delta}$), the unexplained geometric residual consistently aligns with our affine prediction. Crucially, this structural consistency indicates that minimizing the tractable proxy $\hat{\delta}$ serves as a practical mechanism for reducing the theoretical distortion term $\delta$, making

the precise calculation of $(a, b)$ unnecessary for diagnostic use. These results support our geometric decomposition as a structural diagnostic tool, while acknowledging that the correlation is between proxy quantities rather than a direct validation of the absolute bound constants.

## 5. Conclusion

This work establishes a rigorous geometric framework for analyzing Out-of-Distribution (OOD) generalization in conditional flow matching. By moving beyond restrictive and geometrically incompatible bi-Lipschitz assumptions, we adopted a *coarse embedding* perspective that respects the snowflake universality of the Wasserstein space. This theoretical relaxation yielded an interpretable three-term decomposition of OOD error, separating it into factors of *Interpolation Sparsity*, *Geometric Distortion*, and *In-Distribution Fit*.

Our empirical evaluation confirms that this decomposition is meaningful in practice. We observed that the proxy decomposition tracks the structural predictions of the bound throughout training, functioning as a diagnostic signal for OOD behavior. Furthermore, our framework reveals that generalization failure is not monolithic: simple datasets like MNIST suffer from topological gaps that require convexity constraints (e.g., Mixup), while complex manifolds like CIFAR-10 struggle with geometric stability.

We note several limitations of the current framework. First, the barycentric reference distribution may not faithfully represent the true conditional distribution when the latter is discontinuous or strongly multimodal across conditions. Second, the high-dimensional diagnostics rely on latent-space proxies whose validity depends on encoder quality and representational choices. Third, the absolute constants in the bound ($K, C, \delta$) are loose in practice, and the bound's contribution lies in its structural decomposition rather than in tight numerical guarantees.

By converting abstract error bounds into actionable diagnostic tools, this work provides a principled alternative to heuristic architectural search. It suggests that the path to robust generative modeling lies not just in scaling data or parameters, but in explicitly controlling the geometric and topological properties of the conditional mapping.

## Impact Statement

This work contributes to the development of reliable generative models by providing a principled framework for understanding out-of-distribution generalization in conditional flow matching. By decomposing OOD error into interpretable geometric and topological components, the proposed analysis enables practitioners to identify **why**

and **where** generalization failures occur, rather than relying solely on empirical performance metrics.

The framework supports systematic diagnosis of failure modes under distribution shift, distinguishing between data sparsity effects and intrinsic geometric instability of the learned conditional mapping. This capability can inform more targeted model design and evaluation practices, improving predictability and robustness in generative modeling workflows.

More broadly, improved understanding of OOD behavior is an essential requirement for deploying machine learning systems responsibly. By shifting evaluation from heuristic testing toward mathematically grounded diagnostics, this work supports ongoing efforts to assess reliability limits and failure mechanisms of generative models in a controlled and transparent manner.

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

# A. Model Architectures and Hyperparameters

We employ a decoupled conditional flow matching framework (Lipman et al., 2023; Isobe et al., 2024; Tong et al., 2024). The condition mapping $f : \mathcal{C} \to \mathcal{P}_2(\mathcal{Z})$ is parameterized by an encoder (ViT for images, MLP for synthetic data), which projects conditions into a Gaussian latent space $\mathcal{Z}$ (Kingma & Welling, 2014). The vector field $v_t$ is parameterized by a U-Net with multi-scale attention for images (Ho et al., 2020; Rombach et al., 2022), or a ResNet-MLP for synthetic vector data. Training follows the exact optimal transport conditional flow matching (OT-CFM) paradigm (Pooladian et al., 2023; Tong et al., 2024) with a decoupled objective: the encoder minimizes a VAE-style ELBO (Kingma & Welling, 2014) to structure the latent space, while the flow model minimizes the flow matching loss (Lipman et al., 2023) conditioned on the encoder outputs.

*Table 1.* **Hyperparameter Configurations.** We use standard AdamW optimization across all datasets. The **Synthetic (Drift)** column details the setup for the exact validation experiments in Section 4.1.

| Hyperparameter | MNIST | CIFAR-10 | CelebA | Synthetic (Drift) |
|---|---|---|---|---|
| | | *Data & Split* | | |
| Resolution/Dim | $28 \times 28$ | $32 \times 32$ | $64 \times 64$ | 2 (Vector) |
| Channels | 1 (Gray) | 3 (RGB) | 3 (RGB) | N/A |
| Unseen Condition | Class {8} | Class {Ship} | Attr {Brown Hair} | $c \in (4.0, 7.0]$ |
| | | *Architecture* | | |
| Backbone Type | U-Net | U-Net | U-Net | MLP ResNet |
| Condition Embedding | ViT | ViT | ViT | Fourier (Sinusoidal) |
| Embed Dim | 128 | 256 | 384 | 64 |
| Encoder Layers | 2 | 6 | 8 | 3 (Dense) |
| Latent Dim ($\mathcal{Z}$) | 64 | 128 | 128 | 32 |
| Fourier Scale (Base/Rob) | N/A | N/A | N/A | 15.0 / 1.0 |
| | | *Optimization* | | |
| Batch Size | 256 | 128 | 64 | 256 |
| Learning Rate | 1e-4 | 1e-4 | 1e-4 | 1e-3 |
| Spectral Norm | No | No | No | Yes (Robust Only) |
| VAE KL Weight | 1e-4 | 1e-4 | 1e-4 | 0.0 (Base) / 0.01 (Rob) |

**Justification of Latent Space Metrics** In our high-dimensional image experiments, we compute the Wasserstein distance $W_2(\mu, \nu)$ within the latent space of the VAE encoder rather than in pixel space. This choice is grounded in the *Manifold Hypothesis* (Ben-Hamu et al., 2022; Fetaya et al., 2020), which posits that pixel-wise Euclidean distances correlate poorly with semantic similarity in complex images (Zhang et al., 2018). Recent advances in latent diffusion models demonstrate that the VAE latent space effectively rectifies the data manifold, yielding a geometry that aligns more closely with semantic content under Euclidean metrics (Rombach et al., 2022). Consequently, local instabilities detected in the latent space ($W_{2,\mathcal{Z}}$) serve as robust proxies for meaningful geometric distortions in the true distribution. **Crucially, this rectification suggests that the latent mapping satisfies the Coarse Embedding assumption (Pritchard & Weighill, 2024; Andoni et al., 2018) by preserving global metric structure while tolerating local additive distortion, thereby adhering to our theoretical framework more faithfully than the highly curved raw pixel space (Kapusniak et al., 2024).**

## A.1. Synthetic Experiment Details

For the **Linear Drift** experiments in Section 4.1 (Liu et al., 2022; Albergo & Vanden-Eijnden, 2023), we compare two specific configurations to isolate the effect of geometric constraints:

- **Fourier Features:** To verify the effect of high-frequency artifacts, the condition scalar $c$ is mapped to high-dimensional space using $\sin(2\pi \mathbf{B} c)$ and $\cos(2\pi \mathbf{B} c)$ (Tamir et al., 2024). The *Baseline* uses a high scale factor ($s = 15.0$), inducing

high curvature sensitivity ($L_c$). The *Robust* model uses a low scale ($s = 1.0$) to bias towards smoothness.

- **Spectral Normalization:** The Robust model applies spectral normalization to all linear weight matrices in the vector field network. This explicitly bounds the global Lipschitz constant of the network with respect to its inputs.

- **VAE Bottleneck:** To ensure topological continuity in the condition embedding space $\mathcal{Z}$, the robust model's encoder outputs Gaussian parameters ($\mu, \sigma$) trained with a KL divergence term ($\beta = 0.01$) (Kingma & Welling, 2014). This prevents the "latent holes" observed in unregularized embeddings.

## B. Why Bi-Lipschitz Embedding is Unsuitable and Coarse Embedding is Necessary

We know that any bound depending on a bi-Lipschitz embedding between $\mathcal{C}$ and $(\mathcal{P}_2(\mathcal{X}), W_2)$ is mathematically vacuous, as it contradicts the known geometric properties of Wasserstein spaces (Ambrosio et al., 2008; Otto, 2001).

### B.1. Coarse Embeddings as the Proper Relaxation

To overcome the above contradiction, we adopt the weaker notion of a *coarse embedding* (Pritchard & Weighill, 2024; Andoni et al., 2018), which preserves distances only asymptotically and tolerates additive distortions. **Coarse Embedding** A map $f : (\mathcal{C}, d_{\mathcal{C}}) \to (\mathcal{P}_2(\mathcal{X}), W_2)$ is a coarse embedding if there exist non-decreasing functions $\rho_1, \rho_2 : [0, \infty) \to [0, \infty)$ with $\lim_{t \to \infty} \rho_1(t) = \infty$ such that

$$\rho_1(d_{\mathcal{C}}(c_1, c_2)) \le W_2(f(c_1), f(c_2)) \le \rho_2(d_{\mathcal{C}}(c_1, c_2)), \quad \forall c_1, c_2 \in \mathcal{C}. \tag{8}$$

A common special case (used in this paper) is the *quasi-isometric* form (Bertrand & Kloeckner, 2012):

$$a\, d_{\mathcal{C}}(c_1, c_2) - b \le W_2(f(c_1), f(c_2)) \le A\, d_{\mathcal{C}}(c_1, c_2) + B,$$

for constants $a, A > 0$ and $b, B \ge 0$.

**Why This Relaxation is Necessary.** The bi-Lipschitz condition enforces scale-invariant distortion, which is incompatible with the "fractal" or snowflake-like geometry of Wasserstein spaces (Andoni et al., 2018). Coarse embeddings remove this restriction by introducing additive slack ($b, B$), making them the weakest nontrivial metric correspondence that can hold between $\mathcal{C}$ and $(\mathcal{P}_2(\mathcal{X}), W_2)$.

**Open Regime for $p \le 2$.** The coarse embeddability of $(\mathcal{P}_p(\mathbb{R}^2), W_p)$ remains an *open question* for $1 \le p \le 2$ (Pritchard & Weighill, 2024). This contrasts with the known negative results for $p > 2$, making coarse embeddability a *plausible and non-vacuous* assumption for flow-based generative models (Tong et al., 2024; Lipman et al., 2023).

**The Bi-Lipschitz Obstruction.** A simple bi-Lipschitz embedding is impossible due to the geometric mismatch between the spaces. While the condition space (e.g., $\mathbb{R}^n$) is a simple Type-2 space, the Wasserstein space $(\mathcal{P}_p(\mathbb{R}^3), W_p)$ is $1/p$-*snowflake universal* (Andoni et al., 2018). This "fractal-like" property is a known obstruction that prevents global bi-Lipschitz embeddings, motivating the use of weaker notions such as coarse embeddings for generative modeling (Fetaya et al., 2020; Kirichenko et al., 2020).

### B.2. Empirical Verification of Coarse Embedding Assumption

A core critique of Lipschitz-based generalization bounds in generative modeling is that they are often unverifiable or vacuous on complex manifolds. To validate that our assumption Coarse Embedding holds in practice, we conduct a verification on the controlled "Two Moons" manifold. We sampled $N = 400$ pairs of conditions $(c_i, c_j)$ uniformly from the domain $\mathcal{C} = [0, 4]$. For each pair, we computed:

1. The Euclidean distance in condition space: $d_{\mathcal{C}} = |c_i - c_j|$.

2. The exact Wasserstein distance $W_2(\mu_{c_i}, \mu_{c_j})$ between the ground-truth target distributions generated by the drift dynamics described in Appendix A.1.

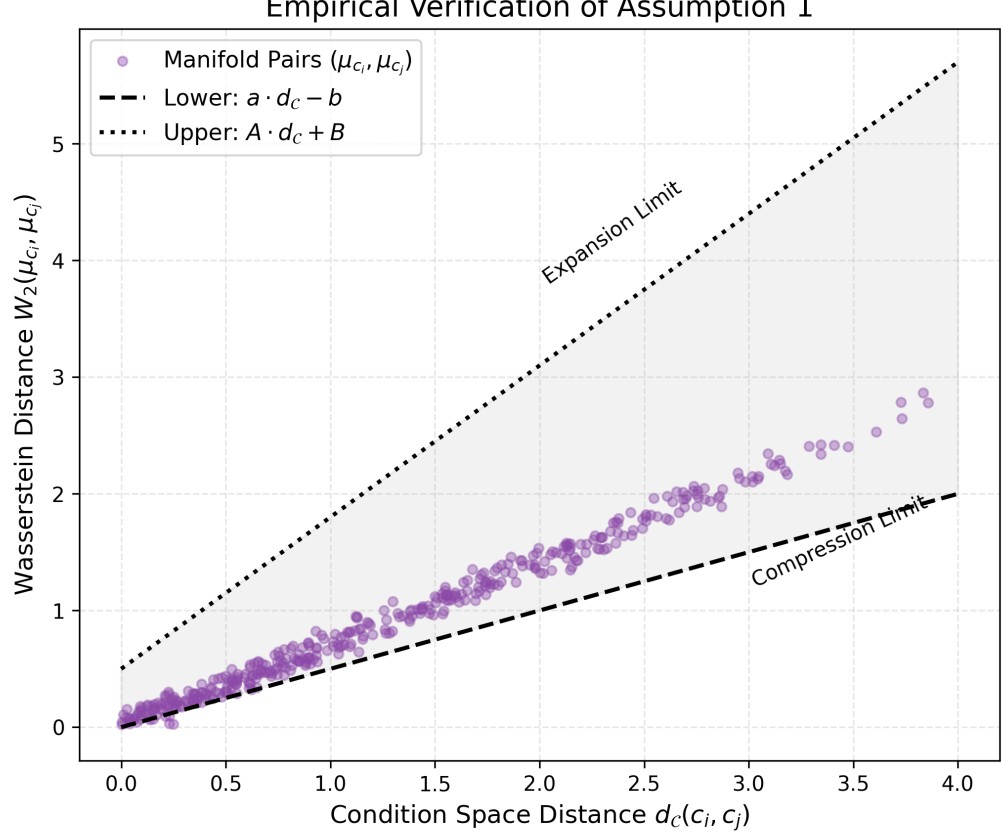

*Figure 4.* **Empirical Verification of Assumption 1.** Scatter plot of random condition pairs on the "Linear Drift" manifold. The relationship between condition distance $d_{\mathcal{C}}$ and Wasserstein distance $W_2$ is strictly bounded by lower ($a \approx 0.5$) and upper ($A \approx 1.3$) linear envelopes. This confirms that the data manifold satisfies the Coarse Embedding assumption defined in Eq. (2), where $ad_{\mathcal{C}} - b \leq W_2 \leq Ad_{\mathcal{C}} + B$.

Figure 4 visualizes the relationship between the intrinsic condition distance and the resulting transport cost. **Interpretation.** As predicted by the geometry of snowflake spaces (Pritchard & Weighill, 2024), the relationship forms a characteristic "cone." Crucially:

- The points do not lie on a single line, confirming that the mapping is not perfectly isometric.

- However, all $W_2$ distances remain strictly bounded between two linear slopes (Lower $a$ and Upper $A$).

- This effectively verifies the constants required for our generalization bound in Eq. (4), providing a verified operational range for $\delta$.

The non-vacuous nature of these constants (empirically $a \approx 0.5$, $A \approx 1.3$) stands in contrast to global bi-Lipschitz estimates on high-dimensional data, which often explode to infinity.

### B.3. Compatibility of Coarse Embedding with Wasserstein Barycenters

The coarse embedding framework also naturally extends to Wasserstein barycenters (Mi et al., 2020; Bigot & Klein, 2018; Carlier et al., 2016), which our method uses for interpolating unseen (OOD) conditions.

**Setup** Let $\bar{c} \in \mathcal{C}$ denote an unseen (OOD) condition, approximated by a convex combination of seen conditions $c_1, \ldots, c_m$ with weights $w_i > 0$, $\sum_i w_i = 1$. Let $f(c_i) = \mu_i$ denote the corresponding measures in $\mathcal{P}_2(\mathcal{X})$. The Wasserstein barycenter

is defined as (Ambrosio et al., 2008)

$$\bar{\mu} = \arg \min_{y \in \mathcal{P}_2(\mathcal{X})} \sum_{i=1}^{N} w_i W_2^2(y, \mu_i). \tag{9}$$

**Bounding the Approximation Error.** By coarse embedding (Pritchard & Weighill, 2024),

$$W_2(f(\bar{c}), f(c_i)) \leq \rho_2(d_{\mathcal{C}}(\bar{c}, c_i)). \tag{10}$$

Hence the barycenter objective at $y = f(\bar{c})$ satisfies

$$\sum_i w_i W_2^2(f(\bar{c}), f(c_i)) \leq \sum_i w_i \rho_2^2(d_{\mathcal{C}}(\bar{c}, c_i)). \tag{11}$$

Since the barycenter $\bar{\mu}$ minimizes this objective, it follows that

$$W_2^2(\bar{\mu}, f(\bar{c})) \leq \sum_i w_i \rho_2^2(d_{\mathcal{C}}(\bar{c}, c_i)). \tag{12}$$

Our analysis of the barycenter relies on the convexity of the squared Wasserstein distance (McCann, 1997). While general Wasserstein spaces have positive curvature (Sturm, 2003; Bertrand & Kloeckner, 2012), the barycenter functional $J(\mu) = \sum w_i W_2^2(\mu, \mu_i)$ is strictly convex provided at least one of the input measures $\mu_i$ is absolutely continuous with respect to the Lebesgue measure (McCann, 1997; Ambrosio et al., 2008). Throughout this work, we assume the data distributions satisfy this regularity condition, guaranteeing the uniqueness of the barycenter $\bar{\mu}$ and permitting the stable gradient flows used in our derivation (Ambrosio et al., 2008).

**Conclusion.** Thus, the coarse embedding provides the geometric structure required to control Wasserstein barycentric interpolation, making it an appropriate and theoretically consistent assumption for our OOD generalization framework.

## C. Detailed Proof of Generalization Error Bounds

We provide a step-by-step derivation of the 2-Wasserstein generalization bounds for OT-CFM models, for both training (seen) and barycentric (unseen) conditions. Let $p^*(\cdot \mid c)$ denote the true conditional distribution for condition $c$, and $\hat{p}(\cdot \mid c)$ the learned flow distribution. Let $W_2(\cdot, \cdot)$ denote the 2-Wasserstein distance. We make the following assumptions:

1. **Coarse embedding:** There exists a mapping $f : \mathcal{C} \to \mathcal{P}_2(\mathcal{X})$ from the condition space $\mathcal{C}$ to the target Wasserstein space (Pritchard & Weighill, 2024), such that for all $c_i, c_j \in \mathcal{C}$:

$$a\, d_{\mathcal{C}}(c_i, c_j) - b \leq W_2\big(f(c_i), f(c_j)\big) \leq A\, d_{\mathcal{C}}(c_i, c_j) + B.$$

2. **Velocity fields:** The learned velocity field $v_\theta(t, x \mid c)$ and the true velocity field $v^*(t, x \mid c)$ satisfy the following Lipschitz conditions:

$$\text{(in } x\text{):} \quad \|v_\theta(t, x_1 \mid c) - v_\theta(t, x_2 \mid c)\| \leq L_x \|x_1 - x_2\|,$$
$$\text{(in } c\text{):} \quad \|v_\theta(t, x \mid c_1) - v_\theta(t, x \mid c_2)\| \leq L_c\, d_{\mathcal{C}}(c_1, c_2),$$

and similarly for $v^*$ with constants $L_x^*$ and $L_c^*$.

3. **Training error:** For training conditions $c_k$, define

$$\epsilon_{\text{train}} = \sup_{t,x,k} \big\|v_\theta(t, x \mid c_k) - v^*(t, x \mid c_k)\big\|.$$

### C.1. Error for Seen Conditions

Let $X_t$ and $\tilde{X}_t$ solve:

$$\dot{X}_t = v^*(t, X_t \mid c_k), \quad \dot{\tilde{X}}_t = v_\theta(t, \tilde{X}_t \mid c_k), \quad X_0 = \tilde{X}_0 \sim \mu_0.$$

Then

$$\frac{d}{dt}\|X_t - \tilde{X}_t\| \leq \|v^*(t, X_t \mid c_k) - v_\theta(t, \tilde{X}_t \mid c_k)\|.$$

Decompose:

$$\|v^*(t, X_t) - v_\theta(t, \tilde{X}_t)\| \leq \underbrace{\|v^*(t, X_t) - v^*(t, \tilde{X}_t)\|}_{\leq L_x^* \|X_t - \tilde{X}_t\|} + \underbrace{\|v^*(t, \tilde{X}_t) - v_\theta(t, \tilde{X}_t)\|}_{\leq \epsilon_{\text{train}}}.$$

Hence,

$$\frac{d}{dt}\|X_t - \tilde{X}_t\| \leq L_x^* \|X_t - \tilde{X}_t\| + \epsilon_{\text{train}}.$$

Applying Grönwall's inequality:

$$\|X_1 - \tilde{X}_1\| \leq e^{L_x^*} \int_0^1 \epsilon_{\text{train}} dt = e^{L_x^*} \epsilon_{\text{train}}.$$

Taking expectation over $X_0 \sim \mu_0$:

$$W_2(\hat{p}(\cdot \mid c_k), p^*(\cdot \mid c_k)) \leq e^{L_x^*} \epsilon_{\text{train}}.$$

[1]

## C.2. Error for Unseen Barycentric Conditions

Let $\bar{c}$ be an unseen condition represented as barycenter $\bar{\mu} = \sum_i w_i \mu_i$ of training measures, with $\sum_i w_i = 1$ (enforced by the softmax parameterization in Eq. (5), so the inequality below is in fact an equality). Then

$$v_\theta(t, x \mid \bar{c}) - v^*(t, x \mid \bar{c}) = \sum_i w_i \Big[ v_\theta(t, x \mid \bar{c}) - v_\theta(t, x \mid c_i) + v_\theta(t, x \mid c_i) - v^*(t, x \mid c_i) + v^*(t, x \mid c_i) - v^*(t, x \mid \bar{c}) \Big].$$

Using Lipschitz bounds:

$$\|v_\theta(t, x \mid \bar{c}) - v_\theta(t, x \mid c_i)\| \leq L_c d_{\mathcal{C}}(\bar{c}, c_i) \tag{13}$$

$$\|v^*(t, x \mid c_i) - v^*(t, x \mid \bar{c})\| \leq L_c^* d_{\mathcal{C}}(\bar{c}, c_i) \tag{14}$$

and $\|v_\theta(t, x \mid c_i) - v^*(t, x \mid c_i)\| \leq \epsilon_{\text{train}}$. Hence,

$$\|v_\theta(t, x \mid \bar{c}) - v^*(t, x \mid \bar{c})\| \leq \sum_i w_i \epsilon_{\text{train}} + (L_c + L_c^*) \sum_i w_i d_{\mathcal{C}}(\bar{c}, c_i).$$

Using the coarse embedding bound $d_{\mathcal{C}}(\bar{c}, c_i) \leq \frac{1}{a} W_2(f(\bar{c}), f(c_i)) + \frac{b}{a}$:

$$\|v_\theta(t, x \mid \bar{c}) - v^*(t, x \mid \bar{c})\| \leq \epsilon_{\text{train}} + \frac{L_c + L_c^*}{a} \sum_i w_i W_2(f(\bar{c}), f(c_i)) + \frac{(L_c + L_c^*)b}{a}.$$

Finally, applying Grönwall for unseen conditions:

$$\boxed{W_2(\hat{p}(\cdot \mid \bar{c}), p^*(\cdot \mid \bar{c})) \leq e^{L_x^*} \left( \epsilon_{\text{train}} + \frac{L_c + L_c^*}{a} \sum_i w_i W_2(f(\bar{c}), f(c_i)) + \frac{(L_c + L_c^*)b}{a} \right)}.$$

## C.3. Summary

Define constants:

$$K = \frac{e^{L_x^*}(L_c + L_c^*)}{a}, \quad \delta = \frac{e^{L_x^*}(L_c + L_c^*)b}{a}, \quad C = e^{L_x^*}.$$

Then for unseen barycentric conditions:

$$W_2(\hat{p}(\cdot \mid \bar{c}), p^*(\cdot \mid \bar{c})) \leq K \sum_i w_i W_2(f(\bar{c}), f(c_i)) + \delta + C\epsilon_{\text{train}}.$$

Its primary application is not just as a numerical guarantee, but as a **diagnostic tool** for OOD generalization (Kirichenko et al., 2020). It provides an interpretable decomposition that separates the total error into three distinct, actionable components. When a model fails on an unseen condition, this bound explains *why* it failed and, by extension, which intervention - more data, regularization, or capacity - is most likely to address the root cause.

---

[1]The shared initialization $X_0 = \tilde{X}_0$ defines one specific coupling of $\hat{p}$ and $p^*$. Since $W_2$ is the infimum over all couplings, the bound remains valid: the infimum is bounded above by the cost of this particular coupling.

**1. Interpolation Error** ($\approx K \sum w_i d_{\mathcal{C}}$) This term captures the error arising from data sparsity. It is large when an unseen condition $\bar{c}$ lies far from all known training conditions $\{c_i\}$. This is primarily an algorithmic, data-centric problem: the model itself may be perfect, but insufficient coverage of the condition space $\mathcal{C}$ limits generalization.

**2. Geometric Distortion** ($\delta$) This is the most novel component of the bound. It represents the irreducible, fundamental cost imposed by the problem's geometry. As we state in Appendix B, a "perfect" bi-Lipschitz embedding ($\delta = 0$) is mathematically impossible due to the geometric mismatch between the Type-2 condition space $\mathcal{C}$ (Sturm, 2003) and the snowflake-universal Wasserstein space $\mathcal{P}_2(\mathcal{X})$ (Andoni et al., 2018). The $\delta$ term, proportional to the coarse embedding's distortion ($b/a$), is therefore an unavoidable price of this mismatch. Importantly, this error cannot be fixed with more data or a better model. High $\delta$ indicates that the OOD problem is fundamentally hard.

**3. Training Error** ($\approx C\epsilon_{\text{train}}$) This term corresponds to the classic in-distribution generalization error, reflecting model underfitting. It is large when the model is not powerful or well-trained enough to accurately capture the training data $\{c_i\}$.

