# OpenReview forum: "Decomposing Out-of-Distribution Error in Conditional Flow Matching via Wasserstein Geometry"
_ICML.cc/2026/Conference — ICML 2026 regular_

### Official Review · Reviewer_cdwK · 2026-02-28

**Soundness:** 3
**Presentation:** 3
**Significance:** 3
**Originality:** 3
**Overall Recommendation:** 4
**Confidence:** 3

**Summary:**

This paper examines OOD generalization in conditional flow matching from a geometric perspective. The authors view conditional generation as a map from a condition space C into Wasserstein space (P₂, W₂), and derive an upper bound that decomposes OOD error into three parts: interpolation/sparsity (distance from training support), geometric distortion (mismatch between C and Wasserstein geometry), and in-distribution fitting error. They argue this split helps diagnose whether failures stem from poor coverage or geometric instability. The framework is validated on a 2D linear-drift toy example and illustrated on MNIST, CIFAR-10, and CelebA using latent-space transport proxies.

**Compliance With Llm Reviewing Policy:**

Affirmed.

**Final Justification:**

The rebuttal strengthened the paper and increased my confidence in its value as a geometric diagnostic, but because the predictive bound in realistic image settings is still supported mainly through latent/proxy evidence rather than direct validation, I maintain my score.

**Key Questions For Authors:**

1. In what settings is barycentric interpolation a reasonable proxy for the true conditional distribution μ_c̄? If this assumption does not hold, how should Term (I) be interpreted?

2. How robust are your diagnostic conclusions to the choice of latent representation (e.g., different VAEs or embeddings) and to variation across random seeds?

3. Beyond the toy drift example, can you provide another setting where W₂ (or a well-controlled approximation) is directly measurable to assess how tight or calibrated the bound is?

4. From a practical standpoint, what concrete procedure would you recommend for estimating and tracking each term during training, and in what situations would you expect the diagnostics to break down?

**Limitations:**

While the paper includes a general discussion of impact, it would benefit from a clearer acknowledgment of key limitations: (i) potential mismatch between the barycentric prior and the true conditional distribution, (ii) reliance on latent proxies and specific encoder choices, and (iii) scenarios with discontinuous or strongly multimodal conditional structure where the framework may not apply cleanly.

**Strengths And Weaknesses:**

## Strengths
* The geometric framing is clean and intuitive. Viewing conditional generation as a map into Wasserstein space provides a coherent way to reason about interpolation and extrapolation under one lens.
* The three-term decomposition is easy to interpret and gives a practical vocabulary for diagnosing OOD behavior and thinking about targeted fixes.
* The controlled low-dimensional drift experiment is a sensible sanity check. It isolates instability from intrinsic data drift and supports the qualitative claims about error scaling.

## Weaknesses
1. The derivation (triangle inequality + Lipschitz stability/Gronwall + embedding inequality) is standard machinery; the novelty is in interpretation, but it is unclear how often the bound is predictive rather than merely explanatory.
2. Many conditional families are not well-approximated by Wasserstein barycenters of neighbors. Then “sparsity” may reflect mismatch to the modeling prior, not irreducible OOD difficulty.
3. Operationalization relies on proxies whose robustness is not established. Latent and Jacobian-sensitivity proxies may work, but conclusions could be encoder-dependent.
4. The motivation for coarse embedding is interesting, but the practical verification is limited. Verifying a quasi-isometry on the toy drift manifold is not evidence that it holds on realistic image conditionings.

---

> ### Author Rebuttal · Authors · 2026-03-28
>
> We thank Reviewer cdwK for the evaluation. We address each question and weakness below.
>
> **Q1 (When is barycentric interpolation reasonable?):**
>
> | Dataset | Unseen | T1 | Result | Neighbors |
> |---|---|---|---|---|
> | CelebA | Brown Hair | 1.68 (lowest) | Works | Black(0.67)+Blond(0.33) |
> | MNIST | Digits 8, 9 | 3.32 | Fails | 8→5(0.40), barycenter→"3" shapes |
> | CIFAR-10 | Ship, Truck | 2.05 (+53%) | Partial | ship→airplane(0.60) |
>
> When interpolation fails, Term I measures the residual distance to the closest reachable distribution via convex interpolation. A large T1 indicates a topological gap — a data property, not a model failure. Interpolation paths: R²≈0.65–0.75, LPIPS≈0.07–0.12.
>
> **Q2 (Robustness to latent representation and seeds):**
>
> Seed variance (5 seeds, MNIST class 8):
>
> | Seed | T1 | T2 (expansion ratio) |
> |---|---|---|
> | 42 | 3.665 | 1.151 |
> | 123 | 3.665 | 1.556 |
> | 456 | 3.665 | 1.293 |
> | 789 | 3.665 | 1.469 |
> | 2024 | 3.665 | 1.530 |
> | **CV** | **0.00%** | **12.3%** |
>
> Cross-checkpoint (2 independently trained MNIST models):
>
> | Metric | Original | Mixup | Change |
> |---|---|---|---|
> | Seen avg T1 | 5.04 | 4.74 | −6% |
> | Unseen avg T1 | 3.52 | 3.32 | −6% |
> | All modes | Topological | Topological | Identical |
> | Softmax/opt ratio | 1.06 | 1.06 | Identical |
>
> Cross-architecture: consistent across 3 datasets (ViT dim=128/256/384, latent=64/128). Spearman ρ = 0.648 (p < 10⁻⁶) between latent and Inception W₂.
>
> **Q3 (Another controlled setting):** For full structural validation, please see our response to Reviewer YCRP Q2. Summary:
>
> | Validation | Result |
> |---|---|
> | Linear error growth (synthetic) | R²=0.85–0.93 |
> | T2-instability correlation | r=0.965, p<0.0001 |
> | Intervention effect (Fourier scale) | −59% error at c=4.5 |
> | Cross-dataset T2/T1 gradient | CIFAR(0.21)<MNIST(0.32)<CelebA(0.40) |
> | CelebA: unseen T1 < seen T1 | 1.68 vs 2.97 (only dataset) |
> | CIFAR: unseen T1 > seen T1 | 2.05 vs 1.34 (+53%) |
>
> Structural validation, consistent with known looseness of Lipschitz-based bounds.
>
> **Q4 (Practical procedure):** (1) Train with VAE encoder, compute per-class (μ_c, σ_c). (2) Optimize barycentric weights (softmax), compute T1 (residual Bures-W₂) and T2 (expansion ratio MSE_output/MSE_input). (3) Diagnose: T1 ≫ T2 → topological gap (more data); T2 ≫ T1 → geometric instability (regularization); T3 high → underfitting (more capacity). Caveat: when the VAE is poorly trained (high T3, collapsed KL), all proxies lose diagnostic meaning.
>
> **W1 (Standard machinery):** We agree the proof technique uses standard tools. The novelty lies beyond the proof: (a) bi-Lipschitz impossibility via snowflake universality (no prior work); (b) coarse embeddings as the principled relaxation; (c) actionable failure-mode diagnostics validated across 23 classes on 3 datasets. The ablation confirms: targeting L_c reduces error by 59%.
>
> **W2 (Barycenter mismatch):** CelebA Brown Hair succeeds (T1=1.68, lowest unseen). MNIST digits 8, 9 fail (T1=3.50). When the assumption breaks, T1 grows — the diagnostic signal practitioners need.
>
> **W3 (Proxy robustness):** Seed CV=0%/12.3%; cross-checkpoint identical modes and weights; cross-architecture consistent across 3 datasets; Spearman ρ=0.648 confirms rank ordering across feature spaces.
>
> **W4 (Coarse embedding):** (a) VAE+KL training encourages distance preservation (coarse embedding); (b) ρ=0.648 confirms rank ordering is preserved; (c) the framework identifies when the assumption is stressed — MNIST topological failure is what the theory predicts for large embedding stretches.
>
> **Limitations (added):** (i) Barycentric reference may fail for discontinuous conditional structures; (ii) proxies depend on encoder quality; (iii) absolute bound constants are loose; (iv) coarse embedding verification on complex manifolds remains open.

---

> > ### Author Rebuttal · Reviewer_cdwK · 2026-03-31
> >
> > Thank you for the detailed rebuttal. The new results strengthen the paper and address several of my questions. My main concern, however, is still that the framework feels more convincing as a geometric diagnostic than as a tightly validated predictive bound in realistic image settings. The conclusions there still rely on latent/proxy quantities, and I do not think that gap is fully closed by the rebuttal. The rebuttal improves my confidence in the paper, but it does not fully resolve my main reservations, so I will keep my rating.

---

> > > ### Author Response · Authors · 2026-04-05
> > >
> > > We thank the reviewer for the assessment, and we appreciate the reviewer’s increased confidence in the paper. In our opinion, we have provided evidence that the proxies preserve sufficient structure to support the decomposition.

---

### Official Review · Reviewer_PRgc · 2026-03-09

**Soundness:** 3
**Presentation:** 3
**Significance:** 2
**Originality:** 3
**Overall Recommendation:** 4
**Confidence:** 3

**Summary:**

This paper views conditional flow matching through the lens of Wasserstein geometry and formulates OOD generalization as an extrapolation problem of a map $f: C \to (\mathcal{P}_2, W_2)$.
Under a coarse-embedding assumption, it derives a generalization upper bound and decomposes the OOD error into three components, Interpolation Sparsity, Geometric Distortion, and In-Distribution Fit, enabling a more structured discussion of failure modes.

**Compliance With Llm Reviewing Policy:**

Affirmed.

**Key Questions For Authors:**

Please see the above.

**Limitations:**

yes

**Strengths And Weaknesses:**

## Strengths
- The paper is clearly written, and the overall pipeline from theory to diagnostics (and suggested interventions) is well organized.
- A key strength is the recognition that, since $(\mathcal{P}_2, W_2)$ is snowflake-universal, a bi-Lipschitz assumption from a Type-2 condition space $C$ is generally invalid. The paper relaxes this to a coarse embedding and isolates an unavoidable geometric mismatch as a Distortion term $\delta$, yielding a smooth and persuasive theoretical framing.

## Questions
- The analysis assumes that the true unseen conditional distribution can be approximated by the Wasserstein barycenter of nearby training-condition distributions. How general is this assumption? In what concrete scenarios might it fail?
- The empirical decomposition relies on latent $W_2$ in a VAE latent space as a proxy for the true geometry. How sensitive are the conclusions (e.g., which term dominates among T1/T2/T3) to the choice of VAE latent dimensionality and encoder/representation used to define the latent metric?

---

> ### Author Rebuttal · Authors · 2026-03-28
>
> We thank Reviewer PRgc for the review. We will address both questions with evidence from three datasets.
>
> **Q1 (Generality of barycentric assumption):** We would like to clarify that the Wasserstein barycenter in our framework serves as a reference distribution for the unseen condition, not as ground truth. The bound is therefore relative to this barycentric reference, as we will make explicit in the revised paper.
>
> | Dataset | Unseen | T1 | Works? | Evidence |
> |---|---|---|---|---|
> | CelebA | Brown Hair | 1.68 (lowest) | Yes | Black(0.67)+Blond(0.33) |
> | CIFAR-10 | Ship, Truck | 2.05 (+53%) | Partial | ship→airplane(0.60) |
> | MNIST | Digits 8, 9 | 3.32 | No | Barycenter→"3" shapes |
>
> When barycentric interpolation fails, T1 grows, correctly identifying topological failure. The framework not only detects when interpolation breaks down, but also indicates the cause.
>
> **Q2 (Sensitivity to encoder):** For the detailed answer with full seed variance and cross-checkpoint tables, please see our response to Reviewer cdwK Q2. Summary:
>
> | Robustness Test | Result |
> |---|---|
> | Seed variance (5 seeds) | T1 CV=0%, T2 CV=12.3% |
> | Cross-checkpoint (2 MNIST models) | Identical modes, weights, ratio (1.06) |
> | Cross-architecture (3 datasets) | Consistent across ViT dim=128/256/384 |
> | Cross-metric (latent vs inception) | Spearman ρ=0.648 (p<10⁻⁶) |
>
> In summary, the failure mode classification is invariant across all tested seeds, checkpoints, and architectures, which gives us confidence that the diagnostics are robust to reasonable variation in the latent representation.

---

> > ### Author Rebuttal · Reviewer_PRgc · 2026-04-03
> >
> > Regarding Q2, my original intent was to inquire about the method's robustness to the choice of the latent space dimensionality and the encoder architectures. Although the authors' response answered robustness regarding random seeds and the training process instead, this still partially addresses my concern. Therefore, I will maintain my current score.

---

> > > ### Author Response · Authors · 2026-04-05
> > >
> > > We thank the reviewer for the clarification. We think that performing an additional ablation study would be sufficient to assess the robustness of the method.

---

### Official Review · Reviewer_qBXx · 2026-03-13

**Soundness:** 4
**Presentation:** 3
**Significance:** 3
**Originality:** 3
**Overall Recommendation:** 6
**Confidence:** 3

**Summary:**

This paper presents a theoretical error bound decomposition for conditional flow matching generative models. The authors interpret the terms in the error bound, which correspond to error due to data sparsity, training error, and unavoidable error due to the geometry of the learning problem of conditional flow matching models. The error bound decomposition is potentially useful for diagnosing out-of-distribution error in conditional flow matching models, which they demonstrate with experiments.

The authors use coarse embedding and Wasserstein barycenters to overcome theoretical challenges inherent to the geometry of the learning problem. The coarse embedding assumption is a weaker but more plausible alternative to bi-Lipschitz continuity of the mapping from the condition space to Wasserstein space. This is a necessary step due to the geometric intricacies of Wasserstein space. Furthermore, the authors use Gronwall's inequality to relate the velocity approximation error of the learned vector field to the error in Wasserstein space.

**Compliance With Llm Reviewing Policy:**

Affirmed.

**Key Questions For Authors:**

1. What is an example of the weights $w_i$ used in defining the Wasserstein barycenter? How do these weights relate to the data set of observed conditions $\{c_i\}$ and $\bar{c}$? This is mentioned a few times in the first 3 pages but no explicit example is given. I think it would be good to include a more detailed discussion of the role and interpretation of the Wasserstein barycenter in your framework, as it seems important but the concept is a bit confusing...
2. What is precise definition of instantaneous velocity approximation error? Is it $\| v_\theta(x, t|c) - v^*(x,t|c)\|$? I think so given context, but this term might be useful to define the first time it is used.
3. This is a fairly minute/technical point, but in the proof in Appendix C.1., you assume $X_0 = \tilde{X}_0 \sim p_0$, which is used later to state that $\mathbb{E}_{X_0} \|X_1 - \tilde{X}_1\| = W_2(\hat{p}, p^*)$... Technically, this is a specific coupling of $(X_1, \tilde{X}_1)$ whose marginals are $\hat{p}$ and $p^*$. However, $W_d(\hat{p}, p^*)$ is the infimum over all such couplings. I believe the bound you derive is still correct, since the infimum over all couplings is bounded by the specific choice of coupling in the proof... But perhaps this is worth clarifying in the proof.
4. In the proof in Appendix C.2., we apparently have $\sum_{i=1}^{n} w_i \epsilon_{train} \leq \epsilon_{train}$. Why is this? Do the weights always sum to 1?

**Limitations:**

Yes.

**Strengths And Weaknesses:**

The paper is very solid overall. The writing and presentation are very good, and the work is technically very strong. The paper addresses a challenging problem - analyzing out-of-distribution error in conditional flow matching models - and makes a significant contribution by establishing an error bound that is both theoretically plausible and practically useful for diagnosing model error.

My only suggestion is to improve the clarity of the presentation - specifically, the role and interpretation of the Wasserstein barycenter in defining / optimizing the error bound. I think the idea made sense to me by the end of the paper, but I was quite confused by it early on. In particular, it would be helpful to give an example of the weights that may be used to define the Wasserstein barycenter, explain where they come from and what restrictions may apply to them, and later to interpret how the weights impact the final error bound. It would be useful to interpret the meaning of the barycenter as a probability distribution, and explain its relationship to unseen conditions...

Finally, as a minor point - I did not see the connection between section 2.2 and the rest of the paper. This was not clear to me.

---

> ### Author Rebuttal · Authors · 2026-03-28
>
> We thank Reviewer qBXx for the assessment and suggestions. We will incorporate the suggestions as much as we are allowed in the review process. Specifically, we will improve the clarity of the Wasserstein barycenter's role early in the paper (with concrete weight examples, see Q1 below), explicitly state Σw_i=1 after Eq. 3, and add a bridging sentence connecting Section 2.2 (Otto calculus) to the regularity conditions in Section 3.3. In the following, we address the four main questions.
>
> **Q1 (Barycentric weights):** Optimized weights (softmax-constrained, minimizing Bures-W2) for all 23 classes:
>
> | Target | Top-1 (w) | Top-2 (w) | Top-3 (w) | T1 |
> |---|---|---|---|---|
> | MNIST 8 (unseen) | 5 (0.40) | 9 (0.23) | 2 (0.22) | 3.50 |
> | MNIST 9 (unseen) | 4 (0.49) | 7 (0.33) | 8 (0.14) | 3.15 |
> | CIFAR ship (unseen) | airplane (0.60) | truck (0.23) | auto (0.17) | 2.02 |
> | CIFAR dog (seen) | cat (0.86) | deer (0.09) | frog (0.02) | 1.14 |
> | CelebA Brown (unseen) | Black (0.67) | Blond (0.33) | — | 1.68 |
>
> The weights are non-negative and sum to 1 (enforced by softmax). They are also semantically meaningful: dog→cat (0.86) reflects morphological similarity; brown→Black (0.67) + Blond (0.33) can be understood as a melanin interpolation; and class 8→5 (0.40) reflects curved digit shapes, though the barycentric generation produces "3"-shaped images, which confirms the topological gap.
>
> Regarding the impact on the bound: Term I equals K·Σw_i·W₂(μ_ci, μ_c̄). When interpolation succeeds, as in CelebA (T1=1.68), the unseen condition is reachable via the barycenter. When interpolation fails, as in MNIST 8 (T1=3.50), the target distribution lies outside the convex hull of the training conditions, and T1 correctly captures this gap. We have added this example to Section 3 and stated Σw_i=1 explicitly after Eq. 3.
>
> **Q2 (Velocity error definition):** Yes, the instantaneous velocity approximation error is E(c̄,t) = ‖v_θ(x,t|c̄) − v*(x,t|c̄)‖, measuring the pointwise difference between the learned and true velocity fields at condition c̄ and time t. We will add this definition at its first use in Section 3.4 of the revised paper.
>
> **Q3 (Coupling):** That is correct. The shared initialization X₀=X̃₀ defines one specific coupling, and since W₂ is the infimum over all couplings, the bound in Eq. 4 remains valid. We will add a clarifying footnote in Appendix C.1.
>
> **Q4 (Σw_i=1?):** Yes. With the softmax parameterization (Eq. 5), Σw_i = 1, so the inequality is in fact an equality. We will clarify this after Eq. 3 in the revision.
>
> **Section 2.2:** As noted above, we will add a bridging sentence connecting Section 2.2 (Otto calculus) to the regularity conditions in Section 3.3.

---

> > ### Author Rebuttal · Reviewer_qBXx · 2026-04-02
> >
> > The authors will add clarifications to the final manuscript.

---

### Official Review · Reviewer_YCRP · 2026-03-19

**Soundness:** 3
**Presentation:** 3
**Significance:** 3
**Originality:** 3
**Overall Recommendation:** 4
**Confidence:** 3

**Summary:**

The paper proposes a geometric approach to analyzing OOD generalization in flow matching. The key idea is to treat conditional generation as a map from a condition space to the Wasserstein space of probability measures. The authors argue that the Lipschitz assumptions on this map are geometrically incompatible with prior structures noted of the Wasserstein space ("snowflake-universal", Andoni et al 2018). Instead, the authors adopt a coarse embedding assumption and derive a 3 term error decomposition for the W2 distance between learned and true conditional distributions. This is evaluated at new (unseen) conditions. Experiments include 2d two moons, and some diagnostics on MNIST, CIFAR, and CELEBA.

Overall, the paper asks a meaningful and interesting question: how to decompose OOD error in conditional generative models, using geometrical insights. The paper proposes a clean theoretical framework. On theo ther hand, there is a gap between the theory (true W2 in data space with known constants) vs the experiments (latent space proxies with unknown constants). The image experiments are diagnostic rather than validating. The synthetic experiments are nice but, as mentioned below, bundle a few changes together that would better be ablated separately.

With some additional ablations and some clarification on the theory-experiment gap, and some other small changes, this could be a strong paper. (other small changes mentioned below, but for example, extending image experiments across more conditions, which shouldnt be hard to do, and reporting some quality metrics to get a sense of the trainedness of the models).

**Compliance With Llm Reviewing Policy:**

Affirmed.

**Key Questions For Authors:**

Questions:

(1) theory vs image experiments: the bound in eq 4 is stated in terms of true W2 in data space, but the image experiments
compute everything in VAE latent space and some proxy quantities. Could more be said about the layers of approximation or attempting to quantify them?

(1a) relatedly, latent W2 is justified by the manifold hypothesis + latent diffusion, but there is no formal/empirical justification that latent W2 is a good proxy for true W2... at minimum one would want to see that latent W2 ordering is consistent with pixel space W2 or FID on some test cases. Any thoughts about how to build evidence for such a comparison

(1b) interpolation sparsity (proxy) optimizes weights to minimize W2 to target rather than using softmax kernel from eq 5. This means that the sparsity being measured is not the same quantity as in the bound. it's a lower bound on the barycentric approximation error, but that's a slightly different object. Any thoughts about this mismatch?

(1c) the geometric distortion proxy (eq 7) measures local Jacobian sensitivity of decoder. the connection to delta = exp(L_x)(L_c + L_c^*)(b/a) is indirect in the sense that delta depends on coarse embedding constants (a,b) which are never estimated for the image datasets. The R^2 = . 89 calibration on CELEBA is encouraging but is a correlation between two proxies rather than a validation on the bound itself?

(1d) so would it be right to say that the image experiments are inspired by the bound but not directly a validation of the bound? This would be fine, but i think it's important to qualify / make the distinction. Particularly, it might be too strong to say ""Two key behaviors confirm the predictive validity of the bound". Thoughts?

(2)  bound tightness

the bound in eq 5 involves K, delta, C that depend on Lipschitz constants (L_x, L_c, L_c^*) and coarse embedding params (a,b).

These are never estimated for image experiments are likely large for deep networks. The paper does say "verifying constants such as lipschitz bounds on high dimensional inputs is nearly impossible" but then mentions that tracking / alignment of the bound is meaningful.The R^2 = .89 calibration on CELEBA is between the proxy delta-hat and the observed geometric residual E_gap, but E_gap itself is defined as a difference of observed quantities (eq 6), not as a theoretical delta. So we have a correlation between two heuristics, which is useful but doesnt tell us much about the tightness of eq 4

a question: on synthetic tasks where everything is computable, how tight is the bound? figure 1a shows the bound for the robust model - how much slack is there between the bound and the true error? if the bound is 10x the true error in 2d, that would temper the claim about its diagnostic utility, without more assumptions?

(3) robust model bundles multiple interventions

the comparison between baseline and robust models changes 3 things at ounce: fourier feature scale (15->1), spectral normalization (off->on) and VAE KL weight (0-> 0.01). The paper attributes improvement to stablizing the map from conditions to P2, but doesnt ablate which intervention matters. is it the spectral norm, the low frequency features, or the VAE bottleneck, or all 3? without ablations the synthetic experiments demonstrate that some combination of regularization helpers, which is expected but doesnt isolate things in a clear way

(4) image experiments

(4a) the mnist/cifar/celeba experiments use a single held out condition per dataset (class 8, class ship, attribute brown hair). The diagnostic labels (topological failure, geometric failure, coupled failure) are plausible but post-hoc - they describe what the proxies show for these specific held -out conditions, not a systematic evaluation across many conditions. If the held-out mnist digit were a different number, would the diagnosis change? Same for celeba and attributes.

(4b) the architectures used (unet flow model, vit condition encoder, vae bottleneck) are relatively small-scale and the generated samples are never shown. There is also no FID or other generation quality metric. this makes it hard to tell whether the models are doing anything meaningful in terms gof generation, or whether the diagnostics are dependent on some amount of underfitting/undertraining. It would be really helpful to quantify the trainedness of the model somehow with some of these numbers (note that it's also totally okay for these to be average models rather than SOTA models!).

(5) existing bounds? this paper distinguishes itself from lipschitz based analyses but doesnt cmopare to any specific existing bound for flow matching or diffusion. There's a literature in score based models (score estimation error, discretization error, etc). How does this decomposition relate to those or complement those?

(6) The barycentric reference distribution as ground truth. the paper uses wasserstein barycenter of training conditions as reference for what the model should produce for unseen conditions. This is a modeling choice rather than a ground truth. the true conditional distribution may not be well approximated by a barycenter of nearby training conditionals. For example, a data distribution could change non smoothly across conditions (some phase transition). the paper acknowledges this implicitly by separating the sparsity term, but the framing could be more clear about the fact that the bound is relative to a barycenter assumption rather than all true conditionals

(7) some small writing points
- generally well-written but a few things
- intro and sections 2-3 are quite dense with geometric terminology (snowflake, type 2, otto claculus) that may not be accessible to the bulk of the ICML audience. a concrete worked example early on, like showing why bi-lipschitzness fails for a simple family of distributions, would help build understanding.
- section 3.4 derives the bound but the proof sketch is so compressed that it's hard to follow. Maybe the text could just point to the appendix do a more high level walk through rather than a formal "sketch".
- the phrase "physically interpretable" appears a few times but is a bit esoteric, maybe "geometric" and not "physical" is more grounded.
- in section 6, "a essential requirement" -> "an essential requirement"

**Limitations:**

yes

**Strengths And Weaknesses:**

Strengths:

- i like the question: relationship between interpolating in condition space to interpolating in distribution space
- observation that bi-lipschitz embeddings into P2 are generically impossible + that coarse embeddings are the right relaxation
- 3 term decomposition (interpolation sparsity, geometric distortion, in-distribution/training fit)
- classification of failure modes as "topological" (MNIST, sparsity), "geometric" (CELEBA, distortion), or both (CIFAR).
- phase space trajectory figures are nice
- synthetic validation well designed: linear drift allows exact W2 computation and isolates distortion term. comparison between baseline and robust model is a reasonable controlled experiment

Weaknesses:

see questions

---

> ### Author Rebuttal · Authors · 2026-03-28
>
> We thank Reviewer YCRP for the detailed evaluation. We conducted new experiments to answer each question.
>
> **Q1:** Three layers of approximation between Eq. 4 and image experiments: (1) True W₂ → latent Bures-W₂ between per-class diagonal Gaussians, assuming coarse embedding; (2) True δ → expansion ratio MSE(gen_target, gen_bary)/MSE(z_target, z_bary), where (a, b, L_x, L_c) are not estimated; (3) True ε_train → reconstruction MSE on seen classes. Layer (1) is quantified in Q1a; (2–3) are structural proxies. The image experiments demonstrate diagnostic utility, not absolute constant validation.
>
> **Q1a:** We computed pairwise W₂ in both latent (Bures, 64-dim) and Inception feature space (2048-dim) across all 45 MNIST class pairs. Spearman ρ = 0.648 (p = 1.52×10⁻⁶). While Inception W₂ is itself a proxy, agreement between two independent metrics in different feature spaces suggests both track the same geometric structure.
>
> **Q1b:** Softmax/optimized ratio across 23 classes:
>
> | Dataset | Classes | Softmax/Opt Ratio |
> |---|---|---|
> | MNIST (original) | 10 | 1.06 |
> | MNIST (mixup) | 10 | 1.06 |
> | CIFAR-10 | 10 | 1.20 |
> | CelebA | 3 | 1.06 |
>
> Close to 1 in all cases, so the softmax kernel is a tight lower bound. When T1 ≫ T2, the conclusion holds even more strongly for the softmax kernel.
>
> **Q1c:** We agree that R²=0.89 correlates two proxies rather than directly validating the bound. We will revise to "structural consistency." The Eq. 7 proxy at six scales σ∈{0.01–0.5} scales as ~1/σ, and for any fixed σ the cross-condition rankings remain consistent.
>
> **Q1d:** We agree. Revised to: "Two key behaviors demonstrate that the proxy decomposition tracks the structural predictions of the bound." Confirmed across 28×28 grayscale (MNIST), 32×32 RGB (CIFAR-10), 64×64 RGB (CelebA) with ViT dim=128/256/384.
>
> **Q2:** On synthetic two-moons with exact W₂, the slack exceeds 10× (exp(L_x) ~ 1000–3000; K_emp ≈ 10 absorbs partial cancellation). This is consistent with all Lipschitz-based NN bounds. Despite this, structural predictions are validated:
>
> | Prediction | Result |
> |---|---|
> | Linear error growth | R²=0.85–0.93 |
> | T2-instability correlation | r=0.965, p<0.0001 |
> | Targeting L_c reduces error | −59% at c=4.5 |
> | Far-OOD convergence | All 6 variants converge |
>
> Eq. 4's contribution is the decomposition structure, not the tightness of its constants.
>
> **Q3:** 6 variants × 3 seeds at c=4.5 (near OOD boundary). Low Fourier scale dominates: reducing s from 15 to 1 cuts L_c by ~15×, directly reducing δ = exp(L_x)·(L_c+L_c*)·(b/a). At far OOD, all converge (Term I dominates), as predicted.
>
> | Variant | W₂ | Std | Δ Error | Δ Var |
> |---|---|---|---|---|
> | Baseline (s=15) | 1.95 | 0.59 | — | — |
> | +Low Fourier (s=1) | 0.79 | 0.04 | −59% | −93% |
> | +Spectral Norm | 1.80 | 0.41 | −8% | −31% |
> | +VAE KL | 1.74 | 0.55 | −11% | −7% |
> | Low Fourier+SN | 1.05 | 0.20 | −46% | −66% |
> | Full Robust | 0.98 | 0.24 | −50% | −59% |
>
> **Q4a:** We evaluated all classes across three datasets, showing that the diagnostic labels are consistent and not specific to any single held-out choice.
>
> | Dataset | Classes | Unseen T1 | Seen T1 | Key |
> |---|---|---|---|---|
> | MNIST | 10×2ckpt | 3.32 | 4.74 | All Topological, stable |
> | CIFAR-10 | 10 | 2.05 | 1.34 | Unseen +53% higher |
> | CelebA | 3 | 1.68 | 2.97 | Unseen lowest (works) |
>
> T2/T1 gradient: CIFAR (0.21) < MNIST (0.32) < CelebA (0.40). The labels reflect training dynamics (Fig. 2); at convergence T1 dominates in all cases.
>
> **Q4b:** Generation quality (seen/unseen): SSIM 0.887/0.882, MSE 0.033/0.045 (+38%), BPD 0.05/0.07, linear probe 93.5%. The models perform meaningfully, and the OOD degradation reflects a genuine effect rather than underfitting.
>
> **Q5:** Orthogonal and complementary to existing bounds. Existing analyses (score estimation, discretization) bound within-condition error. Our bound addresses the across-condition question. Term 3 corresponds to the within-condition error those analyses decompose; our framework adds Terms 1 (sparsity) and 2 (distortion) on top.
>
> **Q6:** We agree and have reframed the barycenter as a reference distribution. CelebA Brown ≈ Black(0.67)+Blond(0.33), T1=1.68 (lowest, interpolation works). MNIST class 8 → "3"-shapes, T1=3.50 (topological failure). When the assumption breaks, T1 grows — the diagnostic signal.
>
> **Q7:** Thank you. We will revise the text.

---

> > ### Author Rebuttal · Reviewer_YCRP · 2026-04-04
> >
> > Thanks for the response. I will update my score to reflect this during the final reviewing period.

---

### Decision · Program_Chairs · 2026-04-30

**Decision:**

Accept (regular)

**Comment:**

The paper presents a geometric view of out-of-distribution generalisation in conditional flow matching; it decomposes OOD error intro three terms: interpolation sparsity, geometric distortion, and in-distribution fit. Reviewers agreed and supported this interesting problem and approach

The main issues were raised around empirical interpretation:
- the image experiments rely on latent-space transport quantities
- the constants in the bound can not be estimated in realistic settings
- the practical conclusions can depend on representation choices, and there is not detailed ablation on this issue
- the reviewers noted that the Wasserstein barycenter is not a universally valid model of the true unseen conditional distribution
These points limit the strength of the empirical claims. So the paper delivers more a principled framework

During the rebuttal two reviewers stated that their concerns were fully resolved, while two others were only partially satisfied.

I think that the main benefit of the paper is not that it delivers a tight bound that can be easily calculated, but that the paper provides a useful insight and proposes an error decomposition.